# Ultrahigh drive current and large selectivity in GeS selector

Shujing Jia [1,2,3,10], Huanglong Li[4,5,10], Tamihiro Gotoh [6], Christophe Longeaud [7], Bin Zhang[8], Juan Lyu[4], Shilong Lv[1], Min Zhu [1✉], Zhitang Song [1✉], Qi Liu [2✉], John Robertson[9] & Ming Liu[2]

Selector devices are indispensable components of large-scale nonvolatile memory and neuromorphic array systems. Besides the conventional silicon transistor, two-terminal ovonic threshold switching device with much higher scalability is currently the most industrially favored selector technology. However, current ovonic threshold switching devices rely heavily on intricate control of material stoichiometry and generally suffer from toxic and complex dopants. Here, we report on a selector with a large drive current density of 34 MA cm$^{-2}$ and a ~10$^6$ high nonlinearity, realized in an environment-friendly and earth-abundant sulfide binary semiconductor, GeS. Both experiments and first-principles calculations reveal Ge pyramid-dominated network and high density of near-valence band trap states in amorphous GeS. The high-drive current capacity is associated with the strong Ge-S covalency and the high nonlinearity could arise from the synergy of the mid-gap traps assisted electronic transition and local Ge-Ge chain growth as well as locally enhanced bond alignment under high electric field.

[1] State Key Laboratory of Functional Materials for Informatics, Shanghai Institute of Micro-System and Information Technology, Chinese Academy of Sciences, Shanghai 200050, China. [2] Key Laboratory of Microelectronic Devices and Integrated Technology, Institute of Microelectronics, Chinese Academy of Sciences, Beijing 100029, China. [3] University of Chinese Academy of Sciences, Beijing 100029, China. [4] Department of Precision Instrument, Tsinghua University, Beijing 100084, China. [5] Chinese Institute for Brain Research, Beijing 102206, China. [6] Department of Physics, Graduate School of Science and Technology, Gunma University, Maebashi 3718510, Japan. [7] Group of Electrical Engineering of Paris, CNRS, Centrale Supelec, Paris Saclay and Sorbonne Universities, Plateau de Moulon, 91190 Gif sur Yvette, France. [8] Analytical and Testing Center of Chongqing University, Chongqing 401331, China. [9] Engineering Department, University of Cambridge, Cambridge CB3 0FA, UK. [10] These authors contributed equally: Shujing Jia, Huanglong Li. ✉email: minzhu@mail.sim.ac.cn; ztsong@mail.sim.ac.cn; liuqi@ime.ac.cn

The remarkable progress of information technology in the twentieth century was powered by the repeated reinvention of the underlying semiconductor devices[1–3]. The advent of random access memory (RAM), hard drive disk (HDD), and Flash memory has stimulated the prevalence of various life-changing consumer electronic devices, such as personal computers and mobile phones[4]. RAM is fast but volatile, low density and expensive, and therefore becomes the primary storage that is the only one accessible to the central processing unit; On the other hand, HDD and Flash are nonvolatile, high density but quite slow, serving as the secondary storage or mass storage[4]. This is known as the computer memory hierarchy[4]. Today, however, the explosive growth of data at a rate of 61% per year, driven by real-time communications and analytics, Internet of Things (IoT) and artificial intelligence, has forced a complete rethinking of the hardware computing baseline from the ground up[5,6]. The base of this stack will be the storage crossover from conventional memories to the new nonvolatile memories that are fast and with large storage capacity[6,7].

Recently, based on the structural phase transition of chalcogenides, three-dimensional (3D) phase change memory (PCM) array has been commercialized by Intel and Micron, which shows the highest density as Flash memory while 1000 times faster (https://www.anandtech.com/show/9541/intel-announces-optane-storage-brand-for-3D-Xpoint-products)[8]. This 3D stacking architecture is enabled by the coupling of a so-called selector element with each memory unit[9]. Selector devices should have ultralow current (OFF current) to avoid any undesired operation of the unselected memory cells; Meantime, they should provide sufficient current (ON current) to drive the targeted one and ensure large read margin[10,11]. In particular, to achieve the Reset operation of PCM cell with 20 nm device size, >20 MA cm$^{-2}$ ON current density ($J_{ON}$) is essential to melt-quench the chalcogenide layer in the PCM[12]. The $J_{ON}$ becomes even larger as the device scales down[12]. Moreover, the ratio of ON current and OFF current, also called as selectivity, should be higher than $10^4$ to realize >Mb capacity[12].

Employing a chalcogenide as PCM and simple sandwich device structure, ovonic threshold switching (OTS) selector seems to be the most suitable candidate[9]. The OTS behavior has been found as early as 1960s[13], and to date, the materials used are either rare Te or Se-based alloy, which often combines with Ge (represented by GeTe$_6$ and GeSe, respectively)[14,15]. To meet the above-mentioned high requirements of PCM arrays, As, Sb, N, and Si elements are commonly incorporated into these alloys to enhance the ON current, increase the selectivity or improve the thermal stability[16–18]. As a result, today's OTS selectors generally contain environment-unfriendly compounds and suffer from material complexity, quaternary or even more. Simple, As-free and stable OTS selectors with outstanding performance are still strongly demanded.

Conventionally, it has been believed that OTS is purely electronic, given its fast switching speed[19]. This has also led to different materials selection criteria from those for PCMs, which undergo a solid to solid-like structural transition. To enable solid to solid-like structural transition, PCM materials are required to be poor glass formers. On the contrary, OTS materials are better glass formers that have slower atomic transition and remain in the amorphous state to higher working temperatures. From an atomic bonding perspective, OTS materials should have stronger bonds, preferably forming saturated covalent (fully connected) networks[20], to survive high currents or high working temperatures, retarding electromigration or the breaking of network bonds. This means using lighter, shorter bond-length elements like Se and S, instead of Te. The degree of covalency can be described by the term-hybridization. As summarized in

hybridization-ionicity map, chalcogen from Te toward lighter S, presenting smaller covalent radius (Te 1.38 Å, Se 1.20 Å, and S 1.05 Å)[21], would lead to larger hybridization when bonding with Ge[22,23]. Noticeably, the hybridization of S-based Ge alloy presents ~15% larger than that of Te-based Ge alloy, enabling large ON current. Moreover, the sharp increase of band gap (GeTe$_6$ 0.6 eV[24], GeSe 1.0 eV[15], GeS 1.5 eV[25]) is beneficial for achieving low OFF current and therefore larger selectivity. As smaller covalent radius also leads to stronger bonds with Ge atom (bond energy: Te-Ge 192 kJ mol$^{-1}$, Se-Ge 234 kJ mol$^{-1}$, S-Ge, 266 kJ mol$^{-1}$), and an increase of the crystallization temperature would be expected to minimize the undesired crystallization. Environment-friendly and earth-abundant S-based alloy is, therefore, an appealing OTS material.

## Results

**Structure and electrical characteristics of GeS device.** Thus, we fabricated GeS-based OTS devices with T-shaped structure by using 130 nm CMOS technology. GeSe-based devices with the same size were also prepared as a reference. Figure 1a shows the schematic structure of the T-shaped device with an Al/TiN/GeS/ W stack. These films were deposited in sequence by RF magnetron sputtering. The 10 nm-thick GeS layer in the device is in its amorphous state, confirmed by the transmission electron microscopy (TEM) image in Fig. 1b. The amorphous GeS can withstand 350 °C for half an hour before any crystallization (Supplementary Fig. 1a), while >450 °C is found for S-rich sample (Supplementary Fig. 1b). As expected, ultralow OFF current, only ~10 nA, passes through the device owing to the large band gap of GeS (Fig. 1c). It then exponentially increases with rising voltage. When the applied voltage reaches $V^{th}$ (threshold voltage), ~3.2 V, the GeS OTS cell turns on and the flowing current suddenly increases to 10 mA (compliance current) by a sharp turn on slope (<2.3 mV dec$^{-1}$, shown in Fig. 1c). The device remains in the ON state until the applied voltage decreases to ~0.5 V ($V_h$, hold voltage). OTS selector generally requires a higher voltage than $V_{th}$ in the first I–V sweep, called firing process, which was not observed in GeS-based cells. Obviously, the drive current is more than one order of magnitude larger than that of GeSe cell with the same device structure (0.5 mA, Supplementary Fig. 2). This means that the $J_{ON}$ of the GeS cell can reach 34 MA cm$^{-2}$, >10 MA cm$^{-2}$ higher than that of Te and Se-based OTS selectors[14,17] to sufficient to drive PCM cells. The ultrahigh ON current (10 mA) and low OFF current (~10 nA) lead to a large selectivity of ~$10^6$. This cell shows excellent reliability in consecutive DC operations (Fig. 1c). For different cells, the ON current remains while the OFF current distributes mainly from ~0.1 to 10 nA, as shown in the statistical distribution of the current in Fig. 1d. The large distribution of OFF current may be due to the rough W bottom surface in our devices. No obvious performance degradation of the selector is found after high temperature annealing (200 °C/30 min and 300 °C/30min, Supplementary Fig. 1c) and 1 h-DC stress test (with ~1/3 $V_{th}$, ~1/2 $V_{th}$, and ~1.3 $V_{th}$ bias, Supplementary Fig. 1d). Moreover, the GeS selector exhibits bidirectional switching behavior, as shown in Fig. 1e, suggesting it can be widely used for other new memory technologies, like resistance RAM (ReRAM). Importantly, as reported in Te/Se-based OTS selectors, the ON/OFF current is size-independent (Fig. 1f and Supplementary Fig. 3a), suggesting larger $J_{ON}$ upon scaling[26]. Since the OTS is a field-induced behavior with a threshold value of ~0.3 V nm$^{-1}$ (Fig. 1f and Supplementary Fig. 3b), the $V_{th}$ is tunable through controlling the GeS film thickness to match the PCM requirements so that it can be read and written successfully. A lower OFF current and higher selectivity is also found as the film becomes thicker.

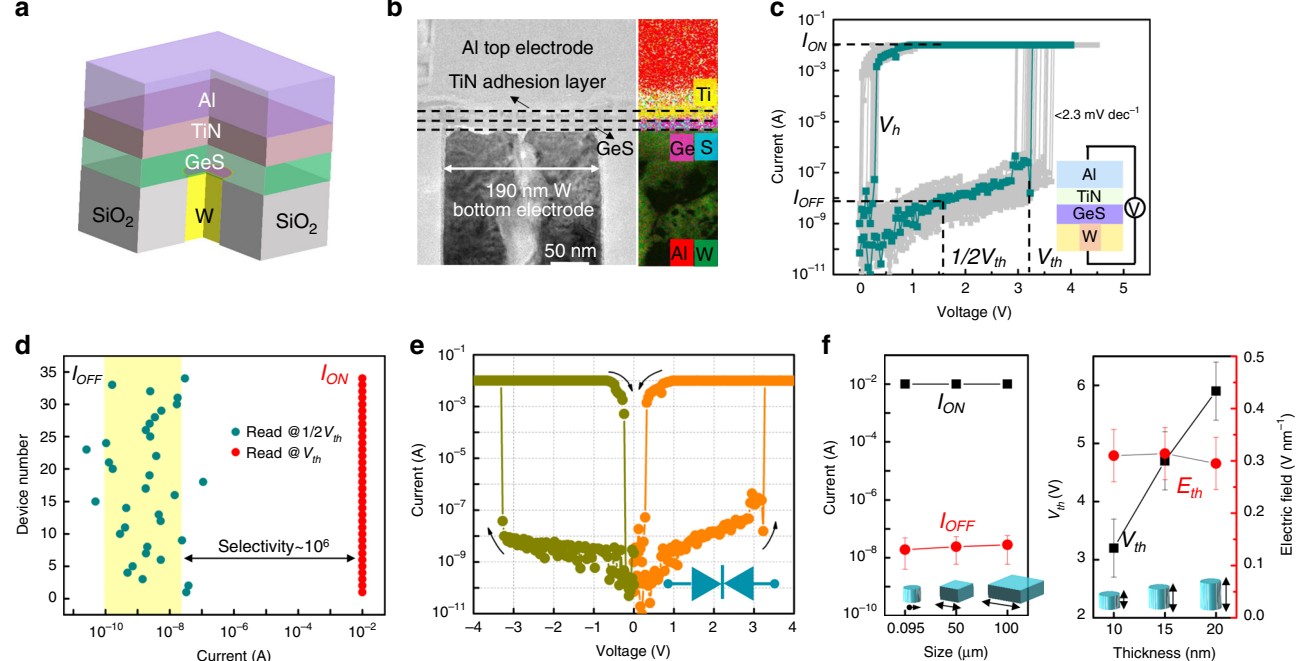

**Fig. 1 Structure and electrical characteristics of GeS device. a** Schematic structure of an individual cell. **b** Cross-section TEM image of device displaying the thickness of each layer and corresponding EDS elemental mapping of W, Ge, S, Ti, and Al. The scale bar is 50 nm. **c** Repeatable DC $I$–$V$ sweeps with uniform compliance current (10 mA) and low leakage current (10 nA). **d** Statistical distribution of OFF current and ON current for various fresh cells measured at $1/2\ V_{th}$ and $V_{th}$, respectively. **e** Bidirectional threshold switching characteristic of GeS device. The arrows represent the switching directions. **f** Device performances with different device sizes and GeS thicknesses. ON/OFF current is independent on the size. The $V_{th}$ linearly increases with increasing thickness, while the electric field is almost unchanged.

To investigate the dynamical transient response of threshold switching, triangular voltage pulses (with 9 V amplitude and 1 μs rising/falling edges) were applied to the GeS selector and a connected resistor, as shown in Fig. 2a. The latter was used to produce transient response curves, which was collected by an oscilloscope. The device turns on within 10 ns at a voltage of ~3.2 V (Fig. 2b), in excellent agreement with $V_{th}$ in Fig. 1c. It then switches back to the OFF state within 100 ns at a voltage of ~0.5 V (Fig. 2c). Almost the same response time is required in multiple triangular pulses operations (Fig. 2d). Although the switching time is longer than Te/Se-based OTS devices[15], it is still comparable with that of conventional $Ge_2Sb_2Te_5$ PCMs (50–100 ns)[8]. Besides, the GeS OTS cells work well with square voltage pulses (Supplementary Fig. 4), implying the ability of timely response to external stimulus in practical application. Figure 2e shows the endurance property of GeS device under square pulses. Up to ~$10^8$ cycles have been achieved with stable ON/OFF current and ~$10^6$ selectivity. The device lifetime is comparable with Te/Se-based OTS selectors (~$10^8$ cycles), and 100 times longer than that of memory cell, >$10^6$ cycles in general[8,18]. Thus, compared to currently available OTS selectors summarized in Fig. 2f[14,17,27–31], GeS ones can provide drive current of >34 MA cm$^{-2}$, which is just smaller than reported B-Te device but ten times larger selectivity. This enables 3D stacking of selector/memory arrays with large PCM operation currents, while having good nonlinearity and reliability. In addition, the used material is a quite simple, binary alloy, which would eliminate the unwanted deviations from the optimized stoichiometry in the wafer scale.

**GeS OTS-based artificial neuron.** Furthermore, these excellent performances also enable the potential use of GeS OTS as artificial neuron with stochastic dynamics behavior[32,33]. The biological neurons functionally integrate input signals received from synapses and create an output one as a threshold is reached, often described as leaky integrate-and-fire (LIF) behavior[32,34,35]. Figure 3a presents the GeS OTS-based neuronal circuit to simulating the LIF behavior, in which parasitic capacitor and series resistance are analogous to the membrane capacitance and axial resistance, respectively. The current response across the GeS OTS device is used as the response spiking signal. Submitted to near-threshold pulse trains, as shown in Fig. 3b, the GeS OTS cell randomly fires after multiple pulses and then relaxes back to the initial state at the end of the pulse trains, resembling to the stochastic dynamics in the biological neurons. Importantly, as the integration time is sharply sensitive to the applied voltage, the firing process of the GeS artificial neuron is tunable by controlling the pulse parameters, like amplitude and rising/falling time. A higher near-threshold square pulse makes the frequency of response faster as shown in Fig. 3c. Under triangular pulse, besides the voltage amplitude (Fig. 3d and Supplementary Fig. 5), the frequency becomes higher as voltage sharply rises as shown in Fig. 3e and f. It significantly increases from 5 to 7 MHz when the rising and falling times are shifted from 8 to 30 μs. The intrinsic large ON current (low resistance) of GeS device makes the parasitic capacitor charge/discharge faster, leading to extremely high-frequency response.

**Band structure of amorphous GeS.** Next, we try to understand why GeS OTS selector exhibits outstanding device performances. First, the microstructure of the GeS device undergoing repeated operations was investigated by high-resolution TEM (HRTEM) and energy dispersive X-ray spectroscopy (EDS), which illustrates the amorphization and homogenous element distribution of GeS film (Supplementary Fig. 6). So, the threshold switching of the GeS device is not caused by the formation/rapture of metal clusters often found in conductive bridge threshold switch (CBTS) selectors[36]. This suggests that the threshold switching of

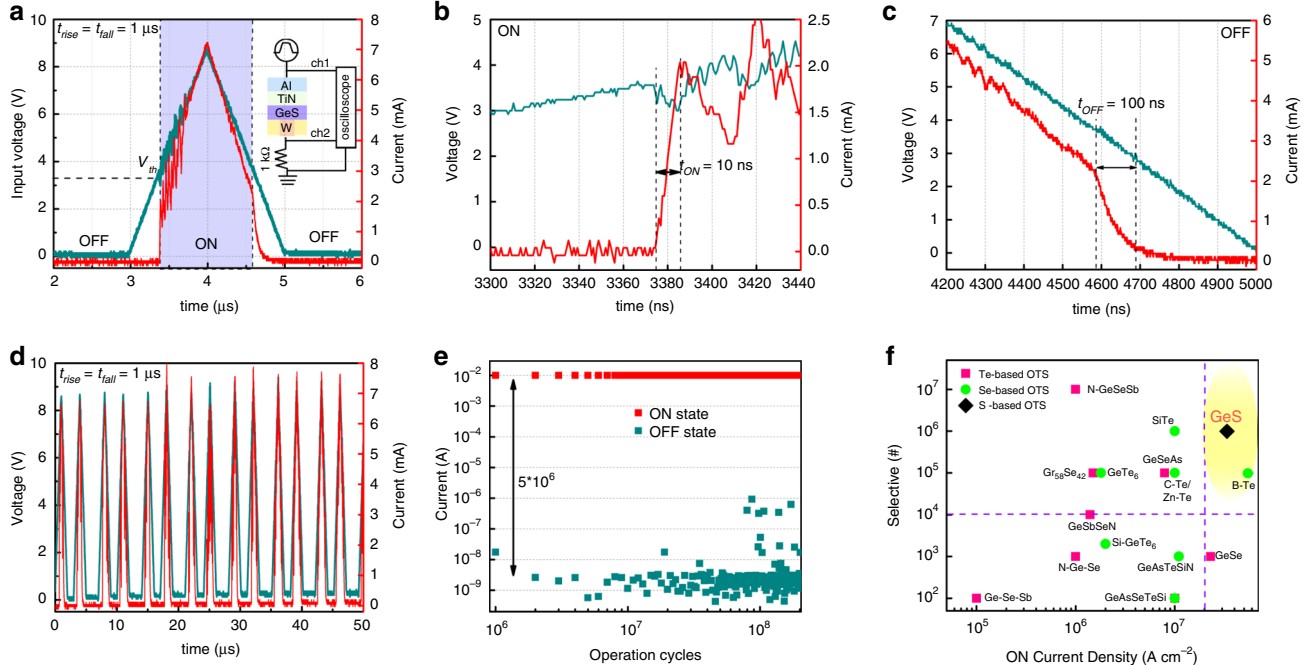

**Fig. 2 Transient characterization and endurance of GeS OTS selector. a** Dynamical transient response of the device under triangular voltage pulses with amplitude of 9 V and rising/falling edge of 1 μs. The device shifts from OFF state to ON state within 10 ns and returns to OFF state within 100 ns. **b, c** Enlarged view of ON and OFF switching curves, respectively. **d** Dynamical response to multiple triangular pulses. **e** Endurance cycles of GeS device maintaining stable high- and low-resistance state. **f** Comparison of selectivity and $J_{ON}$ of reported OTS devices with our work[14,17,27–31]. The ON current density and selectivity for other OTS selectors were calculated from currents and device sizes reported in related works.

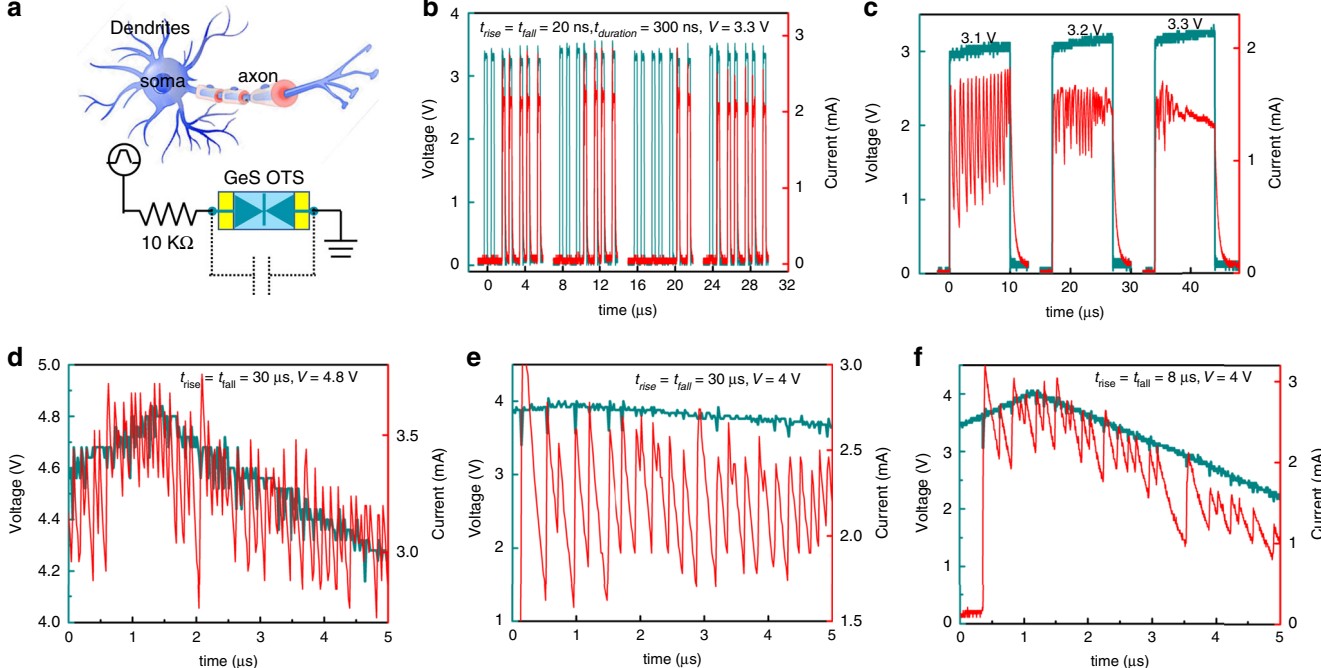

**Fig. 3 GeS OTS-based artificial neuron. a** Schematic of biological neuron and the artificial spiking neuron. A resistor (10 kΩ) is in series with the GeS device and the current response across the device is served as the response spiking signal. Reprinted with permission[35]. Copyright (2012) American Chemical Society. **b** Stochastic behavior of the stimulated neuron applied multiple near-threshold voltage pulses with amplitude (V) of 3.3 V, rising/falling time ($t_{rise}$/$t_{fall}$) of 20 ns, and the duration time ($t_{duration}$), of 300 ns. **c** The response under square pulses with different near-threshold voltages. **d–f** Frequency response of the artificial neuron to which are applied different triangular stimuli pulses, 4.8 V amplitude/30 μs rising/falling time, 4 V amplitude/30 μs rising/falling time and 4 V amplitude/8 μs rising/falling time, respectively.

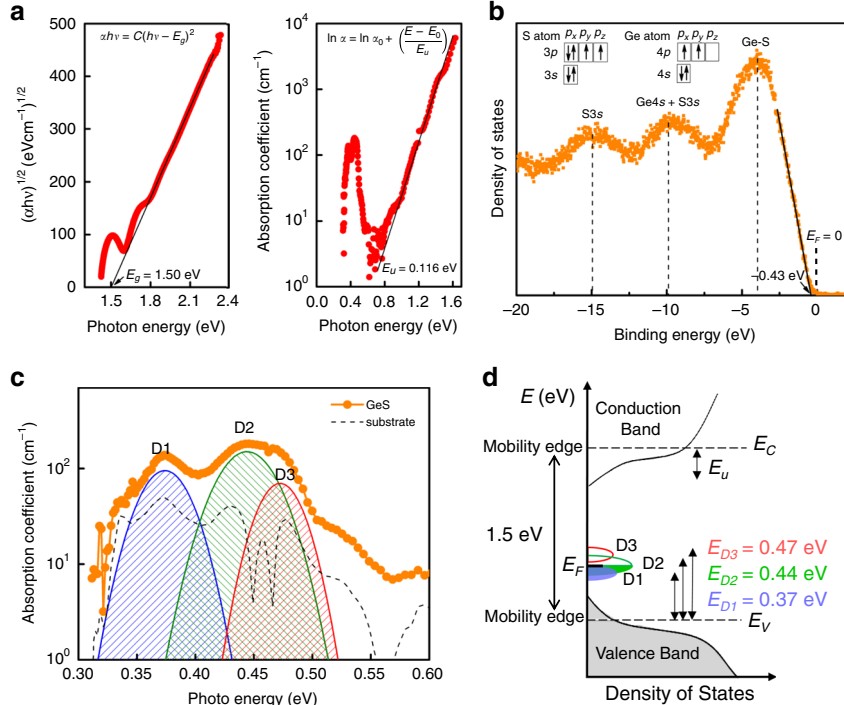

**Fig. 4 Band structure of amorphous GeS. a** Tauc plot of $(\alpha h\nu)^{1/2}$ versus photon energy $h\nu$ of amorphous GeS film indicating the width of the energy gap. The thickness of the GeS film is around 780 nm. Variations of the absorption coefficient with the photon energy revealing the Urbach tail. **b** X-ray excited valence band spectra of GeS film showing the Fermi level. **c** PDS spectrum of GeS film showing the location of localized state. **d** Reconstructed qualitative energy band schematic of GeS glass, where $E_C$ represents the conduction band minimum. Filled traps are those filled with colors, empty traps (without electrons) being left white.

GeS device relies on electronic processes, like Se/Te-based OTS, strongly associated with its energy band structure. The band gap of amorphous GeS can be determined to be 1.50 eV by linear fitting of the near absorption edge in the Tauc plot, $(\alpha h\nu)^{1/2}$ as a function of photon energy $h\nu$ displayed in Fig. 4a, where $\alpha$ is the absorption coefficient, which is consistent with the reported one[25]. A 0.116 eV energy of Urbach tail ($E_u$), known as the width of the localized band tail, can also be obtained by fitting the $\alpha$ versus $h\nu$. Figure 4b shows the X-ray excited valence band spectra of GeS film to estimate the location of Fermi level ($E_F$). The valence band maximum (VBM, $E_V$) is located at −0.43 eV, estimated by a linear extrapolation of the peak to the baseline. Considering the band gap of 1.50 eV, the Fermi level is quite close to the valence band ($E_F - E_V = 0.43$ eV), which gives rise to strongly $p$-type conductivity as reported by Romanyuk[25]. Moreover, density of states (DOS) of outer electrons in the valence band also is shown. The strong peak extending from −7 to 0 eV is mainly associated with the Ge–S bond states rather than reported S $3p$ lone-pair states[37] according to the calculated DOS results.

The trap states within the band gap of amorphous GeS were directly detected by a photothermal deflection spectroscopy (PDS) system extending the energy range down to 0.31 eV (wavelength 4000 nm)[38], as displayed in Fig. 4c. Besides the defect state located at 0.37 eV ($E_{D1}$), two deeper defects located at 0.44 eV ($E_{D2}$) and 0.47 eV ($E_{D3}$) are also found after Gaussian fitting of the broad absorption peak from 0.4 to 0.5 eV. Interestingly, these values are quite close to ~0.45 eV extracted from Poole-Frenkel fitting of temperature dependent $I$–$V$ curves[39] (Supplementary Fig. 7). From absorption peak intensities of PDS, trap densities of D1 to D3 are estimated to be ~$1.2 \times 10^{17}$ cm$^{-3}$ eV$^{-1}$, ~$1.8 \times 10^{17}$ cm$^{-3}$ eV$^{-1}$, and $8.5 \times 10^{16}$ cm$^{-3}$ eV$^{-1}$[40]. They are close to the valence band instead of the conduction band as found in GeSe glass[15]. Combined with the above results, a qualitative band

diagram of amorphous GeS is reconstructed (Fig. 4d), evidencing the relationship between band gap, band tail and traps states. Obviously, $E_F$ is pinned exactly at the D2, which is only enabled by the (electron) filled D1, half-filled D2 and empty D3 since holes are captured by filled states, as illustrated in Fig. 4d. Consequently, the hole carriers excited by the low electrical field (<0.3 V nm$^{-1}$) are frozen into these high-density localized states, resulting in the nA-scale low OFF current.

**Local structure of amorphous GeS.** The identification of structural motifs that may contribute to these trap states can be achieved by Raman measurement and X-ray photoelectron spectroscopy (XPS). As shown in Fig. 5a, the XPS result of Ge $3d$ suggests the coexistence of Ge$^{2+}$ (located at 30.6 eV) and Ge$^{4+}$ (seated at 32.1 eV) states in GeS glass[41]. The Ge$^{2+}$ state is the dominant one (66%), the fraction of which is almost twice as many as the Ge$^{4+}$ one (34%). Unlike Ge, only S$^{2-}$ states are discovered, which are situated at 162.1 eV ($2p_{3/2}$) and 163.3 eV ($2p_{1/2}$). Raman results (normalized) with gaussian fitting are useful for further identifying these motifs (Fig. 5b). The peaks at 210 and 406 cm$^{-1}$ are the vibration modes of the Ge–S chains mostly stacked by Ge-centered pyramids[42]. The former is also observed in the corresponding orthorhombic crystalline structure (used to calibrate the Raman spectra). In contrast, the peak at 291 cm$^{-1}$ is attributed to Ge–Ge bond vibrations in ethane-like S$_3$Ge-GeS$_3$ and GeS$_4$ tetrahedra units with corners occupied by Ge (Ge$_{1+x}$S$_{4-x}$, $1 \le x \le 4$)[42]. Ge–S bond vibrations are also detected at 366 cm$^{-1}$ (in ethane-like S$_3$Ge-GeS$_3$ and edge-shared Ge$_2$S$_6$ tetrahedra)[42]. The Ge atoms in the Ge–S chains are threefold, enabled by the formation of two standard covalent bonds and one coordinate covalent one, as illustrated in Fig. 5c. The standard covalent bond is formed by sharing one $p$ electron from each Ge and S atom. The

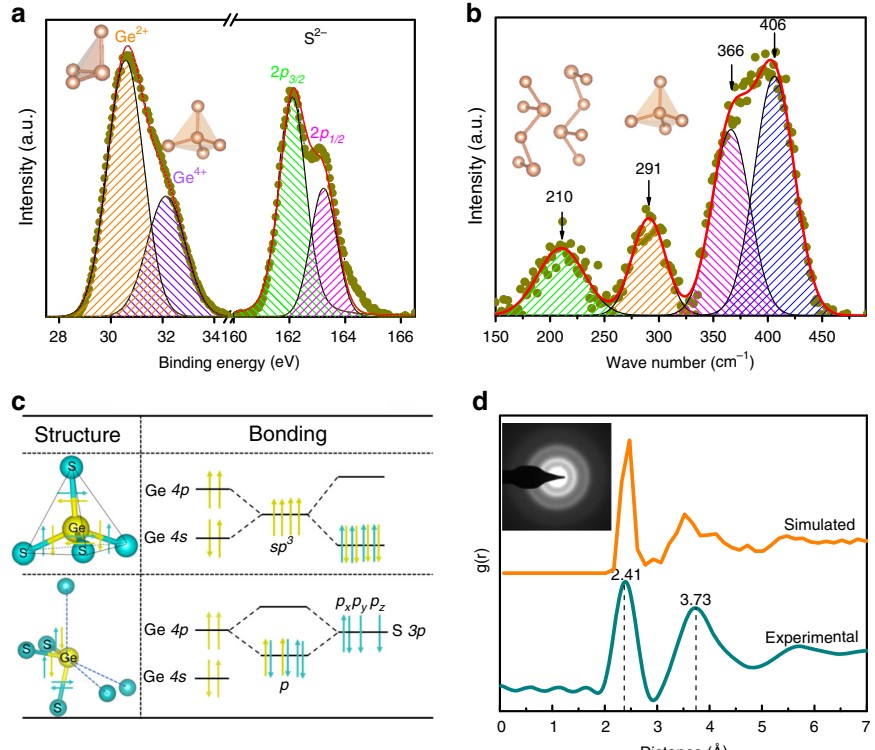

**Fig. 5 Local structures of amorphous GeS. a** The XPS of Ge 3*d* and S 2*p* orbitals. **b** Raman spectrum of GeS film. **c** Atom configuration and bonding environment of pyramidal and tetrahedral Ge-centered structural motifs. **d** Comparison of the experimental RDF by electron diffraction method and simulated RDF obtained by DFT MD simulation. The inset is the corresponding electron diffraction pattern of amorphous GeS film.

left two *p* electrons of S are shared with the empty *p*-orbital of Ge, forming the coordinate bond. In this case, Ge only shares two *p* electrons, which gives rise to the $Ge^{2+}$ state. Besides, twofold Ge can also be attributed to this state. Ge-centered tetrahedra is achieved by $sp^3$ hybridization (Fig. 5c), in which four outer electrons of Ge are fully shared, thereby Ge is in $Ge^{4+}$ state. Ge–S bonds (~2.45 Å length) in threefold pyramidal environment and Ge–Ge bonds (~2.44 Å length) in the tetrahedra have almost the same length[43], both of which contribute to the first peak in radial distribution function (RDF), obtained by electron diffraction experiment of amorphous GeS (Fig. 5d).

**Switching mechanism of GeS selector.** Density functional theory (DFT) calculations were performed to identify the nature of the detected trap states, as well as their roles in the switching process. The structural fidelity of the employed melt-quench-relaxation amorphous GeS network is verified by the good match between the calculated pair radial distribution function and the experimental one (Fig. 5d). The distribution of the coordination numbers is just ~10% different from that found by XPS: 60% of the Ge are threefold coordinated and about 23% are fourfold coordinated, with the rest in twofold coordinated. The amorphous GeS network has a pair of closely spaced Ge atoms, forming interatomic Ge–Ge homopolar bond, and a four-membered Ge chain in which half of the Ge atoms form pyramid structure and the rest forms tetrahedral structure, highlighted by the dashed squares (at regions (1) and (2) of Fig. 6a, respectively). Similar Ge–Ge homopolar bonds and Ge chain motifs have been frequently reported in amorphous GeSe[20,44] and chalcogenide phase change materials[45,46]. The DOS of this amorphous GeS model shows an energy gap of ~1.20 eV (Supplementary Fig. 8), smaller but close to our experimental value (1.50 eV). Filled gap states localized on the Ge pair and the Ge chain are observed (Supplementary Fig. 8). To account for the

experimentally characterized pinning of the Fermi level, empty trap levels are still to be found. To this end, we considered the formation of Ge vacancies because of their acceptor-like property in general. Ge vacancies have been believed to be the origin of high *p*-type conductivity in crystalline GeTe and $Ge_2Sb_2Te_5$ phase change materials[47]. Owing to the hole-doping effect of the Ge vacancy, the Ge pair and Ge chain trap states shift to slightly higher energies, and the Ge chain trap is occupied by hole, resulting in Fermi level pinning between these two trap levels (Fig. 6b), in consistence with the experiment (Fig. 4d). The Ge pair and Ge chain trap states are likely to correspond to those characterized from the PDS (Fig. 4c). The Ge pair and Ge chain trap states provide a ladder for carrier excitation and assist the downward shift of the hole quasi-Fermi level under high electric field[20].

To investigate the atomic and electronic structures of GeS under high electric field, here we simulated the amorphous GeS configuration in the excited state by hole addition considering its *p*-type conductivity. As a result of the excitation, the Ge pair defect state is hole-populated and the mobile hole carriers can appear within the valence band (Fig. 6d and Supplementary Fig. 9). These trap states shift to even higher energies due to further hole doping. Obvious structural changes also occur at the Ge pair as well as at the Ge chain (Fig. 6c): for the Ge pair, one of the Ge atoms become sixfold coordinated, with highly aligned bonds in each of the three directions. This is similar to that observed by Noe et al.[48], where the formation of new metavalent bonds of enhanced alignment has been believed to be the origin of OTS in GeSe. Based on our results, Ge over-coordination is a natural result of the enhanced bond alignment, just as in rock-salt crystalline phase change materials. Unlike the crystallization of phase change materials, the appearance of this sixfold coordinated Ge in GeS is only transient under high electric field excitation. This can be understood from the decreasing tendency

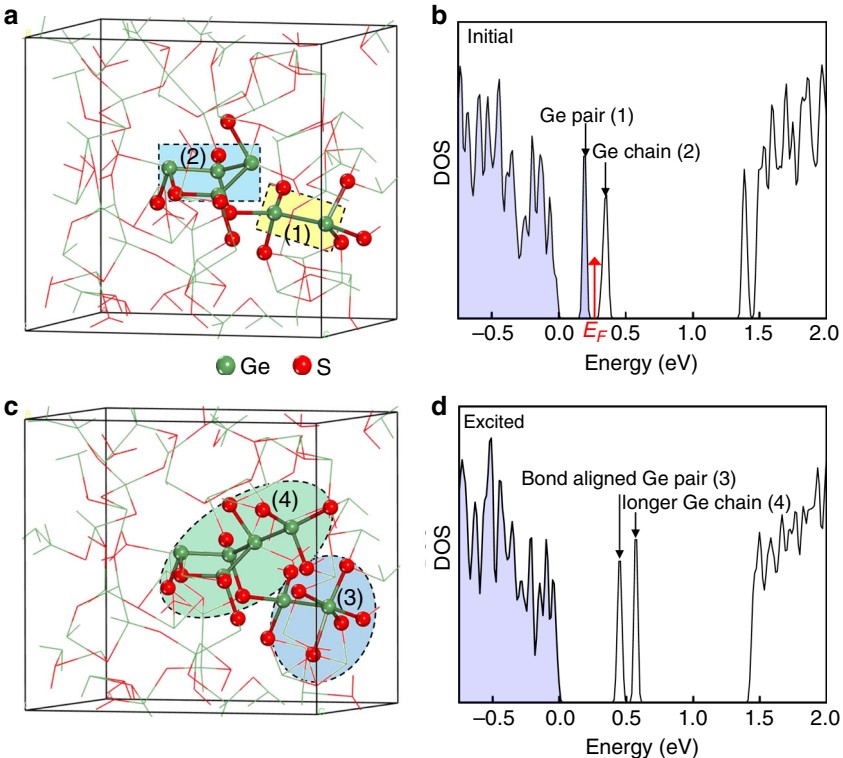

**Fig. 6 First-principles simulation of the switching mechanism.** Evolution of the structure and the corresponding DOS for amorphous GeS (**a**, **b**) before and (**c**, **d**) after hole excitation. The dotted boxes (1) and (2) highlight the Ge pair and Ge chain before excitation, creating one filled and one empty trap state, respectively. The dotted circles (3) and (4) highlight the bond aligned Ge pair and longer Ge chain, respectively.

from GeTe to GeS to form the rock-salt structure (we have compared the differences of the total energies of the orthorhombic structure ($E_o$) and rock-salt structure ($E_r$) for GeTe, GeSe and GeS, the corresponding energy differences ($E_o−E_r$) are 0.59, 0.04, and −0.16 eV, respectively). For the Ge chain, an additional Ge atom joins the chain by forming bond with a Ge in the chain. Guo et al.[49] and Clima et al. also observed the formation of new Ge–Ge bonds and over-coordinated Ge under high field and thermal excitation in GeSe, respectively[45]. The lengthened Ge chain and the more connected network by the formation of over-coordinated Ge could eventually lead to conductive local paths. The electronic transition and structural changes synergistically result in the size-independent high current (ultralow resistivity) of the GeS OTS cell in the ON state (Fig. 1). It should be noted that the system is not simulated under high temperature that may be accessible during the device operation and the carrier excitation takes the Joule heating induced temperature effect into account only indirectly. This simplified approach indeed provides important insight into the mechanism of OTS, but unveiling the materials behavior under an explicit working temperature is still necessary.

Finally, we would like to comment on the possible formation mechanisms of these mid-gap defects. It has been believed that the interactions between the lone-pair (LP) electrons on different chalcogen atoms create the localized gap states[50]. The formation of valence alternation pairs (VAPs) is a common result of these LP interactions because of the low formation energies of VAPs[51]. LP interactions induced gap states, or VAPs induced gap states in particular, have been associated with OTS[19]. This chalcogen LP-based mid-gap defect model works well for systems with chains or clusters of chalcogen atoms, such as amorphous selenium, which have high-lying chalcogen $p$-LP states. However, the presence of chalcogen-chalcogen chains in amorphous germanium

chalcogenides has been questioned experimentally[52] and from DFT calculations[53,54]. In agreement with these works, our experiment and DFT simulations do not seem to support the presence of a significant amount of S–S chains or clusters in the GeS samples. In fact, our simulations indicate that the dominating number of threefold S atoms have deep-lying s-LP states rather than high-lying p-LP states (Supplementary Fig. 10a). Another difference between germanium chalcogenides (GeTe) and other chalcogenide glasses for which the chalcogen LP-based mid-gap defect model works properly has been pointed out to be the high-lying LP electrons being localized on Ge[55]. This still enables VAP formation in germanium chalcogenides[56]. We also plot the isosurface of electron localization function (ELF), which is sensitive to nonbonding electron pairs, for GeS, as shown in Supplementary Fig. 10b. It can be seen that the nonbonding LP orbitals are not only located at S atoms but also at Ge atoms (mainly threefold Ge). A striking feature from our DFT simulated amorphous GeS is the existence of Ge–Ge chain. Indeed, this atomic feature is not exclusive to GeS but seems to be common for germanium chalcogenides, including GeTe and GeSe[44,45,57]. The reason of the existence of Ge–Ge chains has recently been provided with a formation energy using an amorphous GeTe model generated by atomic distortion[58]. Interestingly, the low formation energy of the Ge–Ge chains can be correlated, again, to a VAP formation mechanism, but through interactions between Ge LP electrons. As shown in Fig. 6 and Supplementary Fig. 8, the gap states of GeS are quite localized at the Ge chain and Ge pair structures whose formation is, therefore, also believed to be due to Ge LP interactions. PDOS on S and Ge in amorphous GeS in the excited state (Supplementary Fig. 10c) also shows deep-lying S s-LP states. ELF plot for the excited GeS (Supplementary Fig. 10d) also shows LP orbitals at both S and Ge. The shift of the mid-gap defect energy levels is mainly due to the population change of the

defect levels and the accompanying transient change of atomic structures.

## Discussions

We demonstrate that simple, toxic-free GeS OTS device could yield $34\,MA\,cm^{-2}$ large drive current density and $\sim 10^6$ high nonlinearity for potential 3D stackable memory and neuromorphic applications. Both experiments and first-principles calculations reveal Ge pyramid-dominated network and abundant trap states in amorphous GeS. The high density of localized trap states pins the $E_F$, resulting in low OFF current. These states provide a ladder for carrier excitation and assist the shift of the hole quasi-Fermi level toward valence band under high electric field. In addition, the transient growth of Ge chain and enhancement of local bond alignment enable the high ON current and, thus, high selectivity. This work provides key insights into the unique amorphous atomic structures and electronic band structures of chalcogenides for emerging device applications.

## Methods

**Device preparation and measurement.** T-shaped GeS cells with 190 nm-diameter tungsten plug bottom electrode were obtained by using 130 nm CMOS technology. The 10 nm-thick GeS film was deposited by RF sputtering using GeS alloy target, followed by an adhesion layer of a 10 nm-thick TiN, and a 300 nm-thick Al film (top electrode). The device performances were characterized by a semiconductor device analyzer (Keithley 4200A-SCS).

**Structure and band structure characterization.** A Raman spectroscope (LAB-RAMHR-UV) equipped with a laser with the wavelength of 532 nm was used to obtain the Raman spectra of films at laser power of $\sim 1$ mW avoiding burning out. XPS characterization and the valence spectrum were performed using an ESCA-LAB 250 system equipped with ultraviolet photoelectron spectroscopy accessories with a monochromatized Al K $\alpha$ radiation. The surfaces of the samples were pre-cleaned by $Ar^+$ to remove oxidized and contaminated layers. The Tauc plot and the photothermal deflection spectroscopy was obtained by PDS system extended to a wavelength of 4000 nm at room temperature for samples deposited on quartz glass.

**Cs-corrected TEM characterizations.** The TEM, HRTEM images and EDS analysis were performed with a JEOL JEM-ARM300F microscope operating at 300 kV. The selected area electron diffraction (SAED) of sample of GeS film was performed using Talos F200S microscope and its intensity was integrated using Gatan Diff-tools. TEM samples were prepared by focused ion beam (FIB) using a Helios NanoLab 600 apparatus.

**Atomic simulations.** First-principles calculations were performed by using the plane wave CASTEP code[59]. The electron–electron exchange-correlation energy was defined by the generalized gradient approximation (GGA) functional in the Perdew–Burke–Ernzerhof (PBE) flavor[60]. One-hundred twenty atoms' GeS supercell model was used. Cutoff energy of the plane wave basis set was 380 eV. Corrections to the GGA description of van der Waals bonding were included using the Tkatchenko and Scheffler scheme[61]. The amorphous GeS model was generated using the melt-quench-relaxation method: the structure was first heated at 3000 K for 20 ps and then the liquid phase was quenched from 1000 K to 300 K at a quenching rate of 10 K per picosecond. Considering the aging of the measured glassy GeS samples and OTS devices during which the disappearance of Ge–Ge homopolar bonds is likely to occur, as shown in both OTS[62] and PCM chalcogenides[44,45], we tried to break the long Ge chain in this amorphous GeS model by displacing a Ge atom in its vicinity. The model was further relaxed at 0 K. As a result, the new atomic configuration has a shorter four-membered Ge chain and a dissociated Ge in the center of an S tetrahedron. All atoms were relaxed with gamma-point sampling until the atomic forces on each atom were smaller than $0.02\,eV\,\text{Å}^{-1}$ and the energies are converged to $5\times 10^{-6}$ eV.

## Data availability

All data needed to evaluate the conclusions in the paper are present in the paper and/or the Supplementary Materials. The source data underlying Fig. 1c–e, Fig. 2a–e, and Fig. 3b, d, f are provided as a Source Data file (https://doi.org/10.24435/materialscloud: sc-9m). Additional data related to this paper can be requested from the authors.

## Code availability

Computational results were obtained by using Dassault Systèmes BIOVIA software programs. BIOVIA Materials Studio was used to perform the calculations and to generate the graphical results.

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

## Acknowledgements

Financial support was provided by the National Key Research and Development Program of China (2017YFB0206101, 2017YFB0405601) and Strategic Priority Research Program of the Chinese Academy of Sciences (XDB44010200). M. Zhu acknowledges support by the Hundred Talents Program (Chinese Academy of Sciences) and the Shanghai Pujiang Talent Program (18PJ1411100). J. Robertson acknowledges support by EPSRC (No. EP/P005152/1). H. Li acknowledges support by the National Natural Science Foundation of China (61704096, 61974082), Tsinghua-IDG/McGovern Brain-X program, Youth Elite Scientist Sponsorship (YESS) Program of China Association for Science and Technology (No. 2019QNRC001) and fruitful discussion with Yuzheng Guo and Vaclav Drchal.

## Author contributions

S.J. deposited the film, prepared the devices, and measured device performances. H.L. and J.L. performed the DFT calculations. T.G. and C.L. performed the PDS experiments and analyzed the results. B.Z. extracted the RDF curve from the electron diffraction pattern. S.L. prepared the TEM sample. S.J., H.L., and M.Z. wrote the paper with contributions from T.G., C.L., Q.L., J.R., and M.L. All the authors discussed the results and commented on the manuscript. The project was initiated and conceptualized by M.Z. and Z.S.

## Competing interests

The authors declare no competing interests
