## [Peer Review File · Nature Communications]

Reviewers' Comments:

Reviewer #1:

Remarks to the Author:

The paper investigates GeS materials for OTS applications. It claims record for ON current density, high device selectivity and key insights into the unique amorphous atomic structures and electronic band structures of chalcogenides. GeS materials are not new in memory field but not yet reported in literature for OTS applications.

- Concerning the claimed record for the ON current density, and performances in general, they do not seem the core of the paper. We can find for example in reference "doi:

10.1109/JEDS.2018.2856853" B-Te alloys reported for OTS applications with about 55MA/cm² ON current density. I would not rely on records, but more on device behavior analysis. Being difficult to definitely state on the real formed region (and its surface) in the large device considered in this work, the calculation of the ON current density becomes difficult, also in the light of final given hypothesis about "conductive local paths". In the light of this comment, sentence at line 46 becomes maybe too ambitious.

- Not clear the cumulative distribution reported in Fig.1d. How many devices were tested? The statistic of the results is not reported because of single device testing?

- TEM image of Fig. S6 gives rise to doubts concerning the possible high variability of the device, being high the roughness of the W electrode surface. Being the surface of the device particularly high, how the author can be sure about the no contamination of the GeS layer, which could appear in a localized region of the electrode?

- OTS materials and devices are known for presenting a firing step, needed to "initialize" the device, and to drive it to its stationary behavior. No comments or data are reported here concerning this important aspect. Moreover, all the considerations and explanations concerning the device functionality, should take in account this step.

- The interesting description of the evolution of the GeS system structure leading to the lengthened Ge chains and the more connected network by the formation of over-coordinated Ge, is used to explain the higher material conductivity at high fields, in presence of Ge vacancies. However, the temperature in the "activated" material, in the light of the extremely high current densities proposed, should reach high values. Now, the model proposed takes in account a structural change under "excited state by hole addition", in a solid to solid-like structural transition. How the author comments the capability of the system to sustain such high temperatures, without any structural change? Is this model valid only for the device before firing or even after?

- Fig. 2a shows a 200Mohm resistance at the output of the pulse generator. Is it correct?

Reviewer #2:

Remarks to the Author:

Authors investigated on amorphous GeS environment-friendly and earth-abundant sulfide binary selector material that show large drive current density and meet IRDS standards. Besides, they also showed stochastic integrate-and-fire neuron behavior using GeS device. Most of the experimental and theoretical proofs were conducted almost identically to the GeSe analysis, and are quite reliable. It is considered to be an important paper that can be applied to a future selector device.

1. Please show the error bar according to the repeated experiment for Fig. 1 (f).

2. Authors suggest that it can also be used for new memory technologies such as ReRAM. The switching time (10ns for on, 100ns for off) is longer than Te / Se-based OTS devices. Is this switching time enough to apply to new memory technologies?

3. Please explain the detailed DFT calculation results for S 3p lone-pair states and the effect under high field.

4. Please provide XPS and Raman fitting parameters. From the Fig 5 (a), explain whether the ratio between 2p_{3/2} and 2p_{1/2} from fitting results fits the theoretical values related to spin-orbit splitting. In the Raman spectra, there is information for peaks resulting from the local structure of Ge 4+ and Ge 2+. Please explain how to correlate with the results of 60% 3-fold, 23% 4-fold in XPS.

5. The author described On state as the formation of new metavalent bonds used by GeSe case. Please explain why On-speed is slower than Ge-Se even though Ge-S covalency is very strong.

Reviewer #3:

Remarks to the Author:

S. Jia et al. Prepared and studied a GeS material for the application as OTS selector. The authors could show that GeSe-based OTS selectors yield the best-reported performance for such chalcogenide-based OTS selectors, including device characteristics such as high drive current density and high nonlinearity. In order to understand these advanced properties, the authors studied the local structure of the GeS by theory and experiment. The article is well written and structured. The results are consistent and feasible. I recommend the publication of the submitted manuscript in the journal Nature Communications after some revisions:

1. The reference 1 does not match to the content of the corresponding sentence. The reference 1 reports "a liquid-liquid phase transition in phase-change materials" while the sentence is about "reinvention of the underlying semiconductor devices". I recommend to replace the reference by the following references: Adv. Electron. Mater. 1 (2015) 1400056; Nanoscale Adv. 1 (2019) 3836-3857; Adv. Electron. Mater. 5 (2019) 5; 1900198; Nat. Nanotechnol. (2020).
<https://doi.org/10.1038/s41565-020-0655-z>.

2. Line 101: The Fig. 1b is not a HRTEM image of a device. The authors should provide a HRTEM image of the interface TiN/GeS/W. In addition, the authors should explain the occurrence of bright contrast within of the W bottom electrode. It seems to be porous.

3. Lines 248-250: the authors wrote: "...obtained by electron diffraction experiment of amorphous GeS (Figure 5d), confirm the coexistence of Ge-S bonds (~2.45 Å length) in 3-fold pyramidal environment and Ge-Ge bonds (~2.44 Å length) in the tetrahedral". I would like to point out that the Figure 5d shows the appearance of the first RDF maxima at 2.41 Å, which is consistent with the simulated RDF profile. Thus, RDF analysis could not confirm the "the coexistence of Ge-S bonds (~2.45 Å length) in 3-fold pyramidal environment and Ge-Ge bonds (~2.44 Å length) in the tetrahedral". Consequently, the sentences should be revised and the authors should explain what they see in the RDF analysis. Since TEM is local method, at this point, I would even recommend the authors to perform a RDF analysis based on XRD data in order to see two peaks.

4. For some practical applications, the authors should provide experimental data on thermal stability of GeSe-based OTS selectors where the devices heated up to e.g. 200°C/30 min and 300°C/30min and then record I-V curve at a room temperature (see in DOI 10.1109/LED.2017.2685435).

5. For some practical applications, the authors should provide experimental data on lasting stress test by applying constant voltage to a device (e.g. half and third of threshold voltage) and recording current-time curve (see in DOI 10.1109/LED.2017.2685435).

Reviewer #1 (Remarks to the Author):

The paper investigates GeS materials for OTS applications. It claims record for ON current density, high device selectivity and key insights into the unique amorphous atomic structures and electronic band structures of chalcogenides. GeS materials are not new in memory field but not yet reported in literature for OTS applications.

Reply: We thank you for the thorough review of our manuscript and appreciate your valuable comments. We tried to implement all suggestions and comments to improve the manuscript further.

- Concerning the claimed record for the ON current density, and performances in general, they do not seem the core of the paper. We can find for example in reference “doi: 10.1109/JEDS.2018.2856853” B-Te alloys reported for OTS applications with about 55MA/cm² ON current density. I would not rely on records, but more on device behavior analysis. Being difficult to definitely state on the real formed region (and its surface) in the large device considered in this work, the calculation of the ON current density becomes difficult, also in the light of final given hypothesis about “conductive local paths”. In the light of this comment, sentence at line 46 becomes maybe too ambitious.

Reply: Thank you for your valuable suggestions. Indeed, the purpose of this work is to propose a new GeS OTS material, and then find the underlying mechanism by combing a variety of techniques including electron diffraction technique, XPS, Raman, photothermal deflection spectroscopy and DFT calculations. According to your suggestions, we added and discussed the B-Te OTS result in the Figure 2d, and deleted the term “record” in our manuscript. We also deleted the sentence at line 46 in the abstract.

- Not clear the cumulative distribution reported in Fig.1d. How many devices were tested? The statistic of the results is not reported because of single device testing?

Reply: To show the performance repeatability of GeS-based OTS device, we repeatedly operated the same cell, the I-V curves of which is presented in Figure 1c. The cumulative distribution was obtained from these I-V curves. We also measured many other GeS-based OTS devices (more than 30 cells), the I-V curved are presented in **Figure R1a**. From this result, you can find that GeS-based OTS devices shows large I_{on} with the I_{off} mainly ranging between 0.1 nA to 10 nA, as shown in the cumulative distributions of the current in **Figure R1b**. The large distribution of OFF current may be due to the rough W bottom surface in our devices.

According to your suggestion, we show the repeated I-V curves from one OTS cell and also cumulative distribution of various cells in our manuscript. We replaced Figure 1d by **Figure R1b** in the revised manuscript.

Figure R1 a, Repeatabile DC I-V sweeps obtained from various cells with uniform compliance current (10 mA) and low leakage current (10 nA). The inset shows the schematic diagram of applied cell arrays. b, Cumulative probability of OFF current and ON current for various cells measured at $1/2 V_{th}$ and V_{th} , respectively.

- TEM image of Fig. S6 gives rise to doubts concerning the possible high variability of the device, being high the roughness of the W electrode surface. Being the surface of the device particularly high, how the author can be sure about the no contamination of the GeS layer, which could appear in a localized region of the electrode?

Reply: The used device with 190 nm-diameter W bottom electrodes were obtained by using 130 nm CMOS technology. After deposited the W into the 190 m-diameter hole by chemical vapor deposition (CVD), chemical mechanical polishing was used to remove the W film on the SiO₂ insulator. Since the polish speed of W and SiO₂ are very different, resulting in the rough W electrode surface found in some devices. Not all the devices have this issue.

Before deposited the GeS film on the W electrode, ultrasonic cleaning technique was used to clean these devices. As a result, no contamination of the W was found in the GeS cells, as proved by the EDX mapping of the device in **Figure R2b**. This figure is added in the supplementary materials (**Figure S6**).

Figure R2 a, The HAADF image of GeS-based device. b-e, Corresponding EDS element mappings of W, Ti, Ge and S respectively. f-h, Cross-section HRTEM images. Insets are corresponding Fast Fourier transform images of GeS layer.

- OTS materials and devices are known for presenting a firing step, needed to “initialize” the device, and to drive it to its stationary behavior. No comments or data are reported here concerning this important aspect. Moreover, all the considerations and explanations concerning the device functionality, should take in account this step.

Reply: This is a very interesting point. As your said, many OTS materials, like GeSe, were reported to have a firing process, characterized by much higher required voltage than V_{th} in the first I-V scanning. We also found the firing process in the S-rich GeS OTS device ($Ge_{38}S_{62}$) as shown in the Figure R3. Initially, ~ 9 V voltage was needed to operate the $Ge_{38}S_{62}$ based cell, which then decreased to ~ 5 V in the subsequent operation. Interestingly, no obvious firing process was observed in the Ge-rich GeS OTS device (Ge concentration ≥ 50 at.%). The origin for this phenomenon is still under investigation. **We have added this discussion in our manuscript.**

Figure R3. I-V curves of Ge-S based OTS cells, $Ge_{38}S_{62}$, $Ge_{50}S_{50}$, $Ge_{55}S_{45}$ and $Ge_{70}S_{30}$.

- The interesting description of the evolution of the GeS system structure leading to the lengthened Ge chains and the more connected network by the formation of over-coordinated Ge, is used to explain the higher material conductivity at high fields, in presence of Ge vacancies. However, the temperature in the “activated” material, in the light of the extremely high current densities proposed, should reach high values. Now, the model proposed takes in account a structural change under “excited state by hole addition”, in a solid to solid-like structural transition. How the author comments the capability of the system to sustain such high temperatures, without any structural change? Is this model valid only for the device before firing or even after?

Reply: Thank you for raising these questions regarding the proposed threshold switching mechanism that synergizes both electronic and structural transitions. Conventionally, it has been believed that ovonic threshold switching (OTS) is purely electronic, given its high switching speed. This has also led to different materials selection criteria from those for phase change materials (PCM) which undergo a solid to solid-like structural transition. To enable solid to solid-like structural transition, PCMs are required to be poor glass former. On the contrary, OTS materials are better glass formers that have slower atomic transition and remain in the amorphous state to higher working temperatures. From an atomic bonding perspective, OTS materials should have stronger bonds to survive high currents or high working temperatures, retarding electromigration or the breaking of network bonds. This means using lighter, shorter bond-length elements like Se and S, instead of Te. Due to the stronger atomic bonds, the splitting of the bonding and antibonding states is more significant, giving rise to much larger bandgap values for OTS materials (e.g., GeSe: 1.1 eV, GeS: 1.5 eV) compared with those for PCMs (e.g., a-GeTe: 0.55 eV). In the first-principles simulations, however, we indeed observed transient structural transition upon carrier excitation under high field, indicating that OTS may not be purely electronic but may be assisted by structural changes. Nevertheless, this type of structural transition is only volatile, not permanent; in other words, the network will return to its pristine amorphous state in the absence of driving field. Our proposed mechanism should be more applicable to devices after firing which are cycled by significantly lower voltages than the firing voltages. This ensures that permanent structural changes do not occur as in the firing step.

- Fig. 2a shows a 200Mohm resistance at the output of the pulse generator. Is it correct?

Reply: We appreciate for pointing out this mistake. There is no resistance at the output of the pulse generator. **We have corrected the figure**, which is presented in Figure R4.

Figure R4. Schematic diagram of the dynamical transient response test.

Reviewer #2 (Remarks to the Author):

Authors investigated on amorphous GeS environment-friendly and earth-abundant sulfide binary selector material that show large drive current density and meet IRDS standards. Besides, they also showed stochastic integrate-and-fire neuron behavior using GeS device. Most of the experimental and theoretical proofs were conducted almost identically to the GeSe analysis, and are quite reliable. It is considered to be an important paper that can be applied to a future selector device.

We are very grateful and pleased to read your positive evaluation of our manuscript and the suggestions and comments to further improve it.

1. Please show the error bar according to the repeated experiment for Fig. 1 (f).

Reply: Thank you for this suggestion. We have added the error bar in the Figure 1f, as shown in **Figure R5**.

Figure R5. Device performances with different device sizes and GeS thicknesses.

2. Authors suggest that it can also be used for new memory technologies such as ReRAM. The switching time (10ns for on, 100ns for off) is longer than Te / Se-based OTS devices. Is this switching time enough to apply to new memory technologies?

Reply: For PCM technology, as summarized in the Figure R6 (C. Zambelli *et al.*, Proceedings of The IEEE, 2017, 9, 1790), although GeTe, Sb-doped GST PCM cells can be switched within 50 ns, most of PCM (like GST, N-GST, Ga-Sb-Ge) shows a Set speed of >100 ns. Thus, with 100 ns switching speed, GeS-based OTS cell is fast enough for PCM applications.

In the case of ReRAM, as summarized in the Table R1 (H. Wu *et al.*, Proceedings of The IEEE, 2017, 9, 1770), the typical switching speed of ReRAM is ~50 ns.

Micro-seconds were needed for some ReRAM cells. Therefore, the application of GeS OTS cell in the ReRAM just slightly slow the switching speed.

Figure R6. Correlation between SET time and data retention of PCM devices based on different materials. (C. Zambelli *et al.*, Proceedings of The IEEE, 2017, 9, 1790).

References (affiliation)	Material TE/DM/BE	stack	Cell structure	Storage size and technology node (nm)	Endurance, demonstrated array size (bit)	Retention	Speed and programming conditions
S. Dietrich 2007 JSSC (Qimonda)	TE/Ag/GeSe/BE		1T1R	2M, 90	10 ⁶ , NA	10 ⁴ s, 70°C	Write: 50ns @ 0.6V, Read: 50ns @ 0.15V
C. J. Chevallier 2010 ISSCC (Unity)	NA		1R	64M, 130	NA	NA	Read: 100µs
K. Aratani 2007 IEDM W. Otsuka 2011 ISSCC (Sony)	TE/CuTe/GdO ₂ /BE		1T1R	4K, 180 4M, 180	10 ⁷ , NA	100h, 130°C	Set: 5ns @ 3V, Reset: 1ns @ -1.7V Write: 216Mbit/s, Read: 2.3Gbit/s
S. -S. Sheu 2009 VLSI	TiN/Ti/HfO ₂ /TiN		1T1R	1K, 180	10 ⁶ , NA		Set: 5ns @ 1.8V Reset: 5ns @ -1.6V Read: 8.5ns
Y. S. Chen 2009 IEDM S. -S. Sheu 2011 ISSCC (ITRI)				4M, 180	10 ⁶ , 1K	10yr. 150°C	Switch: 50 ns Read & write: 8ns (single-level), 160ns (multi-level)
M. Wang 2010 VLSI	TaN/Cu ₂ Si ₂ O/Cu		1T1R	1M, 130 8M, 130	10 ⁶ , NA	10yr. 150°C	Set: 2V, Reset: -1V Read: 21ms
X. Y. Xue 2012 VLSI (FDU)	W/Al/PCMO/Pt		1R	1K, NA	NA	NA	Switch: 10µs-10ms @ ±5V
S. Park 2012 IEDM (GIST)	TiN/TiO ₂ /Al ₂ O ₃ /TiN		1T1R	256K, 54	NA	100h, 150°C	Set: 10ns @ 3V Reset: 10ns @ 4V
J. Yi 2011 VLSI	TiN/TiO ₂ /TaO _x /TiN		1R	2M, 54	NA	20h, 150°C	Switch: 10ns, 4V
H. D. Lee 2012 VLSI (Hynix)	NA		1T1R	4M, 65	NA	NA	Write: 0.48-1V, Read: 0.32-1V
M. F. Chang 2012 ISSCC				1M, 28			Set: 500ns @ 25µA, Reset: 100µs @ 50µA, Read: 6.8ns @ 0.85V
M. F. Chang 2014 ISSCC (NTHU)							Write: 0.6V, Read: 0.35V Speed: <100 ns
J. R. Jameson 2013 IEDM N. Gilbert 2013 VLSI (Adesto)	TE/amorphous oxide/BE		1T1R	1Mbit, 130	10 ⁵ , NA	1000h, 200°C	Write: 0.6V, Read: 0.35V Speed: <100 ns
T. Liu 2014 ISSCC (Sandisk)	TE/Metal oxide/BE		1D1R	32G, 24	NA	NA	NA
J. Zahurak 2014 IEDM & S. Sills 2014 VLSI R. Fackenthal 2014 ISSCC (Micron)	TE/Cu-based/BE		1T1R	16G, 27	10 ⁶ , 1M	10yr. 70°C	Write: 180Mbit/s, Read: 900Mbit/s
S. H. Jo 2014 IEDM (Crossbar)	NA		1S1R	4M, 100	10 ⁶ , NA	NA	Selector switching: <50ns
D. C. Sekar 2014 IEDM (Rambus)	TiN/TaOx/HfO ₂ /TiN		1T1R	256K, 120	10 ⁴ , 99% bit yield, 256K	10yr. 70°C	Set: average 14*40ns @ 2.25V
Z. Wei 2008 IEDM	Pt /TaO _x /Pt		1T1R	8K, 180	10 ⁶ , NA	10yr., 85°C	Set: 10ns @ 2V Reset: 10ns @ -1.5V
Z. Wei 2011 IEDM	Ir/Ta ₂ O ₅ /TaO _x /TaN		1T1R	256K, 180	NA	10yr., 85°C	NA
A. Kawahara 2013 ISSCC	Ir/Ta ₂ O ₅ /TaO _x /TaN		1D1R	8M, 180	NA	10yr., 85°C	Write: 443Mbit/s, Read: 25ns
Y. Hayakawa 2015 VLSI (Panasonic)	Ir/Ta ₂ O ₅ /TaO _x /TaN		1T1R	2M, 40	10 ⁵ , 2 M	10yr., 85°C	NA
H. W. Pan 2015 IEDM (NTHU)	TE/HfO ₂ -based/BE		1FinFET1R	1K, 16	10 ⁵ , NA	1000h, 150°C	Set: 50ns @ 2.3V, Reset: 1ms @ 1.8V
M. Ueki 2015 VLSI (Renesas)	W/Metal/Ta ₂ O ₅ /Ru		1T1R	2M, 90	NA	40min, 200°C	Write: 2.5V Read: 0.5V
X. Huang 2015 IMW (THU)	TiN/TaOx/HfO ₂ /TiN		1T1R	1K, 120	10 ⁶ , single cell; 10 ⁵ , 99.7% bit yield, 1K	10 ⁴ s, 85°C	Set: 50ns @ 1.75V, Reset: 50ns @ 1.9V

Table R1. Summary of RRAM properties with different materials (H. Wu *et al.*, Proceedings of The IEEE, 2017, 9, 1770).

3. Please explain the detailed DFT calculation results for S 3p lone-pair states and the effect under high field.

Reply: Thank you for suggesting an investigation of the S 3p lone-pair states. It has been believed that the interactions between the lone-pair (LP) electrons on different chalcogen atoms create the localized gap states (S. R. Ovshinsky and K. Sapru, Taylor

& Francis, London, 1974, p. 447). The formation of valence alternation pairs (VAP) is a common result of these LP interactions because of the low formation energies of VAPs (M. Kastner *et al.*, *Phys. Rev. Lett.*, 1976, 37, 1504). LP interactions induced gap states, or VAPs induced gap states in particular, have been associated with the ovonic threshold switching (OTS) phenomena (D. Adler *et al.*, *J. Appl. Phys.*, 1980, 51, 3289). This chalcogen LP-based midgap defect model works well for systems with chains or clusters of chalcogen atoms, such as amorphous selenium, which have high-lying chalcogen p-LP states. However, the presence of chalcogen-chalcogen chains in amorphous germanium chalcogenides has been questioned experimentally (P. Jovari *et al.*, *Phys. Rev. B*, 2008, 77, 035202) and from density-functional-theory (DFT) calculations (S. Caravati *et al.*, *Appl. Phys. Lett.*, 2007, 91, 171906). In agreement with these works, our experiment and DFT simulations do not seem to support the presence of a significant amount of S-S chains or clusters in the GeS samples. In fact, our simulations indicate that the dominating number of three-fold S atoms have deep-lying s-LP states rather than high-lying p-LP states (figure R7b). Another difference between germanium chalcogenides (GeTe) and other chalcogenide glasses for which the chalcogen LP-based mid-gap defect model works properly has been pointed out to be the high-lying LP electrons being localized on Ge [A. V. Kolobov *et al.*, *Phys. Rev. B*, 2013, 87, 155204]. This still enables VAP formation in germanium chalcogenides [A. V. Kolobov, *Sci. Rep.*, 2015, 5, 13698]. We also plot the isosurface of electron localization function (ELF), which is sensitive to nonbonding electron pairs, for GeS, as shown in figure R7b. It can be seen that the nonbonding LP orbitals are not only located at S atoms but also at Ge atoms (mainly 3-fold Ge). A striking feature from our DFT simulated amorphous GeS is the existence of Ge-Ge chain. Indeed, this atomic feature is not exclusive to GeS but seems to be common for germanium chalcogenides, including GeTe and GeSe [Phys. Rev. B 92, 054201 (2015), Phys. Rev. B 93, 115201 (2016), Nat. Commun. 6, 7467 (2015), Sci. Rep. 9, 1867 (2019), Microelectron. Eng. 215, 110996 (2019)]. The reason of the existence of Ge-Ge chains has recently been provided with a formation energy explanation by Li and Robertson [H. Li and J. Robertson, *Appl. Phys. Lett.*, 2020, 116, 052103] using an amorphous GeTe model generated by atomic distortion. Interestingly, the low formation energy of the Ge-Ge chains can be correlated, again, to a VAP formation mechanism, but through interactions between Ge LP electrons. As shown in Figure 6 and Figure S8, the gap states of GeS are quite localized at the Ge chain and Ge pair structures whose formation is therefore also believed to be due to Ge LP interactions.

Figure R7 a, c Projected density of states (PDOS) on S and Ge of amorphous GeS in the ground state and excited state, respectively. b, d Isosurface of electron localization function of amorphous GeS in the ground state and excited state, respectively

PDOS on S and Ge in amorphous GeS in the excited state also shows deep-lying S s-LP states. ELF plot for the excited GeS (figure R7d) also shows LP orbitals at both S and Ge. The shift of the mid-gap defect energy levels is mainly due to the population change of the defect levels and the accompanying transient change of atomic structures.

This discussion is added in the revised manuscript and Figure R7 was shown in the supplementary information as Figure S10.

4. Please provide XPS and Raman fitting parameters. From the Fig 5 (a), explain whether the ratio between $2p_{3/2}$ and $2p_{1/2}$ from fitting results fits the theoretical values related to spin-orbit splitting. In the Raman spectra, there is information for peaks resulting from the local structure of Ge $4+$ and Ge $2+$. Please explain how to correlate with the results of 60% 3-fold, 23% 4-fold in XPS.

Reply: Thank you for this valuable suggestion. Indeed, the fitting parameters for XPS and Raman did not match well with the theoretical values. We have re-fitting them. The fitting parameters of XPS and Raman results are summarized in **Table R2** and **Table R3**. Now the ratio between $2p_{3/2}$ and $2p_{1/2}$ peaks from fitting results is ~ 2 , matching well with the theoretical values related to spin-orbit splitting. The re-fitting results are show in **Figure 5a and b** in the revised manuscript (**Figure R8a and b**).

	Peak	Position(eV)	Area	FWHM(eV)
Ge	1	30.6	21106.2	1.6
	2	32.1	10553.1	1.7
S	1	163.3	7316.0	1.1
	2	162.1	14632.0	1.1

Table R2. The fitting parameters of XPS spectra.

Peak	Position (eV)	Area	FWHM (eV)
1	210.3	2.7	51.0
2	290.8	2.8	36.0
3	366.2	6.0	41.1
4	406.1	8.1	43.3

Table R3. The fitting parameters of Raman spectra.

Figure R8 a, The XPS and corresponding fitting result of Ge 3d and S 2p orbitals. b, Raman and corresponding fitting spectra of GeS film.

The results of 60% 3-fold, 23% 4-fold were obtained from melt-quench-relaxation amorphous GeS network. The concentration of 4-fold motif is ~10% lower than that found in the XPS. In the XPS and corresponding results, the fraction of Ge⁴⁺ one (4-fold) is ~34%, almost half of that of Ge²⁺ state (3-fold+2-fold). These structural motifs also can be detected by the Raman results, as shown in Figure R8a. From the fitting result in **Table R3**, we can find that 4-fold motif is the dominated one with higher area ratio. However, we cannot get the exact fraction because the unclear relationship of the different peak area and correspond structural fractions in Raman spectra.

5. The author described On state as the formation of new metavalent bonds used by GeSe case. Please explain why On-speed is slower than Ge-Se even though Ge-S covalency is very strong.

Reply: Thank you for raising another interesting point. The On state of GeS-based OTS is enabled by the synergy of the mid-gap traps assisted electronic transition and local Ge-Ge chain growth as well as locally enhanced bond alignment under high electric field. The transient growth of Ge-Ge chain needs to break the dominated Ge-S bonds and then adjust the local structure. The stronger Ge-S bonds than Ge-Se bonds

means that the local structure is more stable and cannot be easily changed. This results in the relatively slower On-speed of GeS based OTS cell in our work.

Reviewer #3 (Remarks to the Author):

S. Jia et al. Prepared and studied a GeS material for the application as OTS selector. The authors could show that GeSe-based OTS selectors yield the best-reported performance for such chalcogenide-based OTS selectors, including device characteristics such as high drive current density and high nonlinearity. In order to understand these advanced properties, the authors studied the local structure of the GeS by theory and experiment. The article is well written and structured. The results are consistent and feasible. I recommend the publication of the submitted manuscript in the journal Nature Communications after some revisions:

We thank you for your detailed review and the encouraging evaluation of our manuscript. We tried to implement all suggestions and comments to improve the manuscript further.

1. The reference 1 does not match to the content of the corresponding sentence. The reference 1 reports “a liquid-liquid phase transition in phase-change materials” while the sentence is about “reinvention of the underlying semiconductor devices”. I recommend to replace the reference by the following references: *Adv. Electron. Mater.* 1 (2015) 1400056; *Nanoscale Adv.* 1 (2019) 3836-3857; *Adv. Electron. Mater.* 5 (2019) 5; 1900198; *Nat. Nanotechnol.* (2020). <https://doi.org/10.1038/s41565-020-0655-z>.

Reply: Thank you for recommending these references. They are cited in our manuscript for replacing reference 1.

2. Line 101: The Fig. 1b is not a HRTEM image of a device. The authors should provide a HRTEM image of the interface TiN/GeS/W. In addition, the authors should explain the occurrence of bright contrast within of the W bottom electrode. It seems to be porous.

Reply: Thank you for pointing out this mistake. Figure 1b was a TEM image of a device. We have corrected it. The HRTEM image of the interface TiN/GeS/W is shown in **Figure R9**.

This figure was added in the supplementary information (Figure S6).

Figure R9 a, The HAADF image of GeS-based device. b-e, Corresponding EDS element mappings of W, Ti, Ge and S respectively. f-h, Cross-section HRTEM images. Insets are corresponding Fast Fourier transform images of GeS layer.

Indeed, as you said, there is a porous within the W bottom electrode. The depth of the W bottom electrode is ~ 530 nm, as shown in the **Figure R10**, which was deposited by chemical vapor deposition. Since it is so deep that sometimes holes appear inside the W bottom electrode. Not all the devices have this issue.

Figure R10 Correctional TEM image of our device.

3. Lines 248-250: the authors wrote: “...obtained by electron diffraction experiment of amorphous GeS (Figure 5d), confirm the coexistence of Ge-S bonds (~ 2.45 Å length) in 3-fold pyramidal environment and Ge-Ge bonds (~ 2.44 Å length) in the tetrahedral”. I would like to point out that the Figure 5d shows the appearance of the first RDF maxima at 2.41 Å, which is consistent with the simulated RDF profile. Thus, RDF analysis could not confirm the “the coexistence of Ge-S bonds (~ 2.45 Å length) in 3-fold pyramidal environment and Ge-Ge bonds (~ 2.44 Å length) in the

tetrahedral". Consequently, the sentences should be revised and the authors should explain what they see in the RDF analysis. Since TEM is a local method, at this point, I would even recommend the authors to perform a RDF analysis based on XRD data in order to see two peaks.

Reply: Thank you for another valuable suggestion. We have revised the sentences: Ge-S bonds (~2.45 Å length) in 3-fold pyramidal environment and Ge-Ge bonds (~2.44 Å length) in the tetrahedra have almost the same length⁴⁰, both of which contribute to the first peak in radial distribution function (RDF), obtained by electron diffraction experiment of amorphous GeS (Figure 5d). The RDF result is also used to verify the structural fidelity of employed amorphous GeS network in the discussion part.

Also, since the bond lengths of Ge-S bonds and Ge-Ge are almost the same, 2.45 Å and 2.44 Å, respectively, X-ray diffraction technique cannot distinguish them, as reported by N. Fueki *et al.* (**Figure R11**, Journal of the physical society of Japan, 1992, 6, 2814).

Figure R11 Total pair distribution functions of amorphous Ge-S systems obtained X-ray diffraction technique (N. Fueki *et al.*, Journal of the physical society of Japan, 1992, 6, 2814).

4. For some practical applications, the authors should provide experimental data on thermal stability of GeSe-based OTS selectors where the devices heated up to e.g. 200°C/30 min and 300°C/30min and then record I-V curve at a room temperature (see in DOI 10.1109/LED.2017.2685435).

Reply: Thank you for suggesting an investigation of GeS-based cell under different temperatures. We annealed two devices at 200 °C and 300 °C for 30 min, respectively, and then measured their I-V curve at the room temperature. The results are presented

in **Figure R12**. Although the V_{th} and V_{hold} are a ~ 0.5 V different, no obvious performance degradation of the selector is found.

Figure R12 I-V curves of GeS-based OTS cells annealed at 200 °C/30 min and 300 °C/30min, respectively.

Figure R12 are shown in the **Figure S1(c)** in the supplementary information.

5. For some practical applications, the authors should provide experimental data on lasting stress test by applying constant voltage to a device (e.g. half and third of threshold voltage) and recording current-time curve (see in DOI 10.1109/LED.2017.2685435).

Reply: According to your suggestion, we have performance DC stress test of GeS-based OTS cell by applying 1 V ($\sim 1/3 V_{th}$), 1.5 V ($\sim 1/2 V_{th}$) and 4 V ($\sim 1.3 V_{th}$) bias, as shown in **Figure R13**. No obvious degradation of ON and OFF performances is observed.

Figure R13 DC stress test of GeS-based OTS cell by applying 1 V ($1/3 V_{th}$), 1.5 V ($1/2 V_{th}$) and 4 V ($1.3 V_{th}$) bias.

Figure R13 are shown in the **Figure S1(d)** in the supplementary information.

Reviewers' Comments:

Reviewer #1:

Remarks to the Author:

Thank you for the careful revision of the manuscript and answering to the several questions raised. The work is interesting for the community and I recommend the publication.

There are still some points I would like to report to the author:

- In caption of Fig.2f, since ON current density is a difficult parameter to extract, and this is why not always reported in papers, I warmly suggest to add at the end of the caption a sentence similar to the following: "maximum ON current density and selectivity for other OTS selectors, were estimated from currents and device sizes available, when not clearly reported in previous works."
- Even in revised Fig.1d, now reporting data for different devices, y-axis represents a cumulative probability (%) going from 0 up to 35%. This is not a common way to plot the data. From my understanding, y-axis should have as definition "device number" instead of "cumulative probability". It is worthwhile also to add if ON and OFF currents reported for the 34 cells are for fresh or cycled devices (and in this last case how many cycles).
- The answer to the 5th question is not complete, and the proposed "thermal excitation" used often in DFT also for the study of other chalcogenide compounds to compensate the impossibility to simulate a real high electric field, limits the understanding to what could happen at the switching event supposing that electric field is generating such "induced" excitation, but without taking in account the huge temperature increase in the system due to the extremely high current density involved. The description of the structure during the application of such high current density remains unrevealed. The temperatures you are dealing with are probably in the range of the melting one (not that high for GeS₂), while your assumption that "OTS materials... remain in the amorphous state to higher working temperatures" implies that the material should deal with "solid-to-solid" transition. This is still a subject of debate, I believe that the analysis of the device after 1E8 cycles will provide some input about that, revealing degradations that are on the contrary due to such extremely high localized temperature increase.

Reviewer #2:

Remarks to the Author:

We thank the authors for faithfully answering all the answers. All answers were logically well answered, so it is recommended that the paper be published in nature communications.

Reviewer #3:

Remarks to the Author:

The authors revised their manuscript according to my suggestions. My recommendation is the acceptance of the revised manuscript and its publication in the present form the Journal Nature Communications

REVIEWER COMMENTS

Reviewer #1 (Remarks to the Author):

Thank you for the careful revision of the manuscript and answering to the several questions raised. The work is interesting for the community and I recommend the publication.

Reply: We thank you for the careful review of our manuscript again and appreciate your valuable comments. Thanks also for recommending the publication of our work in Nature Communications.

There are still some points I would like to report to the author:

- In caption of Fig.2f, since ON current density is a difficult parameter to extract, and this is why not always reported in papers, I warmly suggest to add at the end of the caption a sentence similar to the following: “maximum ON current density and selectivity for other OTS selectors, were estimated from currents and device sizes available, when not clearly reported in previous works.”

Reply: Thanks for your suggestion. We added the sentence in the end of the caption of Fig. 2f.

- Even in revised Fig.1d, now reporting data for different devices, y-axis represents a cumulative probability (%) going from 0 up to 35%. This is not a common way to plot the data. From my understanding, y-axis should have as definition “device number” instead of “cumulative probability”. It is worthwhile also to add if ON and OFF currents reported for the 34 cells are for fresh or cycled devices (and in this last case how many cycles).

Reply: Thanks for this valuable suggestion. We replaced the “cumulative probability” by “device number” in Fig. 1d.

- The answer to the 5th question is not complete, and the proposed “thermal excitation” used often in DFT also for the study of other chalcogenide compounds to compensate the impossibility to simulate a real high electric field, limits the understanding to what could happen at the switching event supposing that electric field is generating such “induced” excitation, but without taking in account the huge temperature increase in the system due to the extremely high current density involved. The description of the structure during the application of such high current density remains unrevealed. The temperatures you are dealing with are probably in the range of the melting one (not that high for GeS₂), while your assumption that “OTS materials... remain in the amorphous state to higher working temperatures” implies that the material should deal with “solid-to-solid” transition. This is still a subject of debate, I believe that the analysis of the device after 1E8 cycles will provide some input about that, revealing degradations that are on the contrary due to such extremely high localized temperature increase.

Reply: Thanks for raising this question. As you have pointed out, the increase of the temperature, caused by the high current density, cannot be avoided. According to your suggestion, we have analyzed the GeS OTS device after repeated operations, the TEM and EDX results are shown in

Figure R1. Clearly, although high current has had passed through the device, the GeS film remained in the amorphous structure after that (Figure R1 f-h). This means that the thermal accumulation was still insufficient to induce the crystallization of GeS (Figure R2a). Based on these results, it is reasonable to ignore the thermally driven permanent solid-to-solid transition in our DFT calculations before any performance degradation occurs (less than 1E8-cycle operations).

After 1E8-cycle operations, the OTS device performance began to deteriorate (Figure 2e), which indeed suggested the change of the structure or composition in the GeS film. This may be due to the gradual diffusion after long-time operation or high electrical field-induced electromigration. We cannot agree more that the analysis of device failure mechanism is important; however, it is another research subject which cannot be fully addressed in this manuscript focusing on the switching mechanism of GeS based OTS device. Nevertheless, we would like to underscore the our philosophy of choosing GeS as the OTS material with respect to the minimization of permanent solid-to-solid structural changes: to avoid permanent solid to solid-like structural transition, OTS materials should be better glass formers that have slower atomic transition and remain in the amorphous state to higher working temperatures, in contrast to PCM materials such as GeTe. From an atomic bonding perspective, OTS materials should have stronger bonds, preferably forming saturated covalent (fully connected) networks, to survive high currents or high working temperatures, retarding electromigration or the breaking of network bonds. This means using lighter, shorter bond-length elements like Se and S, instead of Te.

Thanks for your consideration.

Figure R1 and Figure R2 are shown in Figure S6 and Figure S1 in the supplementary information, respectively.

Figure R1 | a, The HAADF image of GeS-based device. b-e, Corresponding EDS element

mappings of W, Ti, Ge and S respectively. f-h, Cross-section HRTEM images. Insets are Fast Fourier transform image of GeS layer. This GeS device has undergone repeated electrical operations. Homogenous element distributions of GeS film, without metal filament, are observed in the repeatedly operated selectors, as shown in Figure S6 a-e. Also, the GeS film maintains its amorphous state (Figure S6 f-h), which proves that the threshold switching of these OTS selectors is the result of electronic processes, unlike the Conductive bridge threshold switch (CBTS) and Phase change memory (PCM).

Fig. R2 | XRD patterns of annealed a, GeS and b, Ge₃₈S₆₂ films at different temperatures. 200 nm-thick GeS and Ge₃₈S₆₂ were deposited on SiO₂/Si substrate annealed at different temperatures for 0.5 h. A small amount of GeS crystalline phase appears in the film annealed at 400 °C, while the sample at 350 °C is completely amorphous. Hence, the crystallization temperature is higher than 350 °C, which is higher than reported mature selection material systems-GeSe. A crystallization temperature higher than 450 °C is observed for Ge₃₈S₆₂.

Reviewer #2 (Remarks to the Author):

We thank the authors for faithfully answering all the answers. All answers were logically well answered, so it is recommended that the paper be published in nature communications.

Reply: We appreciate your recommendation for the publication of our work in Nature Communications.

Reviewer #3 (Remarks to the Author):

The authors revised their manuscript according to my suggestions. My recommendation is the acceptance of the revised manuscript and its publication in the present form the Journal Nature Communications.

Reply: We appreciate your recommendation for the publication of our work in Nature Communications.

Reviewers' Comments:

Reviewer #1:

Remarks to the Author:

Dear Author,

Thank you for the corrections and your careful review.

I already recommended the paper for publication.

Concerning the last question, I think my point has been misunderstood. I try to explain it better. The simulations performed, as you confirmed, do not take in account the real temperature reached in the system during the ON state, and you agreed on the fact that such temperature should be really high: "... the excited state by hole addition" is not sufficient for justifying that the studied structure is compatible with the one achieved in the real ON state of the device. I agree with your proposals and hypothesis to justify why OTS devices could survive to high currents and high temperatures, but I can't find this answer in the proposed simulations.

I suggest to add at the end of your discussion, for fairness, that the model and the simulations proposed, compatibly with the others proposed in the literature that you quoted, can give an insight into the mechanisms involved in the switching of GeS materials without taking in account the temperature increase, however ON state structure should be better investigate in the light of future results and findings.

I hope to find your agreement on this open, really interesting, scientific aspect.